# Colorectal adenoma presence is associated with decreased menaquinone pathway functions in the gut microbiome of patients undergoing routine colonoscopy

Ilona Vilkoite[1,2]*, Ivars Silamiķelis[3], Jānis Kloviņš[3], Ivars Tolmanis[4], Aivars Lejnieks[5,6], Elīna Runce[7], Krista Cēbere[8], Ksenija Margole[9], Olga Sjomina[4,8,10], Laila Silamiķele[3]

**1** Department of Doctoral Studies, Riga Stradins University, Riga, Latvia, **2** Health Centre 4, Riga, Latvia, **3** Latvian Biomedical Research and Study Centre, Riga, Latvia, **4** Digestive Diseases Centre GASTRO, Riga, Latvia, **5** Department of Internal Diseases, Riga Stradins University, Riga, Latvia, **6** Riga East University Hospital, Riga, Latvia, **7** Faculty of Medicine, Riga Stradins University, Riga, Latvia, **8** Residency in Gastroenterology, Riga Stradins University, Riga, Latvia, **9** Residency in Family Medicine, Riga Stradins University, Riga, Latvia, **10** Institute of Clinical and Preventive Medicine, University of Latvia, Riga, Latvia

\* ilona153@inbox.lv

## Abstract

### Background

Colorectal adenomas are key precancerous lesions and a major target for colorectal cancer prevention. While gut microbiome alterations are well described in colorectal cancer, microbial composition and functional capacity at the adenoma stage remain poorly understood. Emerging metagenomic data suggest early adenomas are associated with loss of microbial metabolic functions supporting epithelial and immune homeostasis.

### Objectives

To investigate the association between gut microbiome composition and functional pathways and the presence of colorectal adenomas in patients undergoing routine colonoscopy.

### Materials and methods

This cross-sectional case–control study included adult patients undergoing routine colonoscopy. Participants were enrolled based on strict inclusion and exclusion criteria to minimize confounding factors such as inflammatory bowel disease, prior colorectal surgery, and recent antibiotic or probiotic use. Fecal samples were collected prior to bowel preparation, and gut microbiome taxonomic composition and functional pathways were analyzed using shotgun metagenomic sequencing.

**Data availability statement:** All metagenome sequencing files are available from the European Nucleotide Archive (accession number PRJEB79034) https://www.ebi.ac.uk/ena/browser/view/PRJEB79034.

**Funding:** This work was funded by the European Regional Development Fund (ERDF), Measure 1.1.1.1 "Support for applied research" Project No.1.1.1.1/18/A/092 „Role of miRNAs in the host-gut microbiome communication during metformin treatment in the context of metabolic disorders".

**Competing interests:** The authors have declared that no competing interests exist.

## Results

A total of 136 participants were included, of whom 56 had colorectal adenomas. Alpha diversity indices did not differ significantly between adenoma-positive and adenoma-negative groups. In contrast, beta diversity analysis revealed significant differences in overall microbial community structure. Descriptive genus-level differences suggested features of dysbiosis in adenoma-positive patients, including higher relative abundance of *Bacteroides* and *Prevotella* and lower abundance of *Faecalibacterium* and *Anaerostipes*. Differential abundance analysis identified a single species-level feature, UBA7597 sp003448195, enriched in the adenoma group. Functional profiling showed reduced microbial pathways related to menaquinone (vitamin $K_2$) biosynthesis, Stickland fermentation, and short-chain fatty acid (propionate) production in patients with adenomas.

## Conclusions

The presence of colorectal adenomas was associated with reduced microbial metabolic functions linked to vitamin $K_2$ biosynthesis, amino acid fermentation, and propionate production, alongside compositional shifts toward a less functionally robust gut microbiome. These findings indicate that early colorectal neoplasia is accompanied by functional microbiome alterations that may serve as markers of adenoma-associated dysbiosis and provide insight into early metabolic changes in the colonic microenvironment.

## Introduction

Colorectal cancer (CRC) remains a significant global health challenge, representing a substantial burden on both healthcare systems and affected individuals [1]. Owing to its high incidence and mortality rates, CRC ranks among the most prevalent and deadly malignancies worldwide. According to recent epidemiological data, CRC accounts for approximately 10% of all cancer cases and is the third most commonly diagnosed cancer globally, with an estimated 1.8 million new cases diagnosed annually [1].

The etiology of CRC is multifaceted, involving a complex interplay of genetic, environmental, and lifestyle factors [2]. Major risk factors include high consumption of red meat coupled with low fiber intake, obesity, lack of physical activity, substance abuse, and persistent bowel inflammation [3]. Among these factors, emerging research has increasingly recognized the pivotal role of the gut microbiome in the development and progression of CRC [4–6]. The human colon harbors a diverse and dynamic community of microorganisms, collectively known as the gut microbiota, which plays a fundamental role in maintaining intestinal homeostasis and influencing various aspects of host physiology [7]. Typically, polyps progress to malignancy following a well-defined path known as the adenoma-carcinoma sequence [8]. Alternatively, 15–30% of CRCs develop through the serrated pathway [9].

Polyps in the premalignant stage from both pathways can be screened and removed during colonoscopy to prevent the formation of CRC. However, if polyps are incompletely removed or go undetected, this can lead to the emergence of interval cancers. The formation of colorectal polyps precedes the development of cancer and is impacted by a range of environmental factors and the host's genetics [8].

Alterations in the composition and function of the colon microbiota, including, but not limited to, dysbiosis have been implicated in the pathogenesis of CRC, particularly when these changes disrupt immune homeostasis, barrier integrity, or metabolic balance. Mounting evidence suggests that dysbiotic changes in the gut microbiota can contribute to chronic inflammation, aberrant immune responses, and metabolic dysregulation within the colonic microenvironment, thereby promoting tumorigenesis and tumor progression [10]. Understanding the intricate relationship between the colon microbiota and CRC pathogenesis holds immense promise for developing novel preventive and therapeutic strategies for this devastating disease.

Growing evidence suggests that qualitative or quantitative changes in the abundance of specific gut microbiota members could serve as markers for the future development of colorectal neoplasia. Although various studies have explored changes in gut microbiota composition in the context of colorectal adenomas, the results remain inconclusive [11–13], highlighting the need for further research. Specifically, published studies differ in the reported direction and magnitude of microbial diversity changes, as well as in the identification of taxa associated with adenoma presence. Moreover, methodological heterogeneity, including differences in sequencing platforms, analytical pipelines, lesion characteristics, and population backgrounds, has contributed to inconsistent and sometimes conflicting findings.

Recent large-scale metagenomic studies have advanced understanding of the gut microbiome's role in the early stages of colorectal carcinogenesis. Lee et al. [14] analyzed stool metagenomes from 971 colonoscopy-screened participants, identifying distinct microbial and functional profiles associated with tubular and sessile serrated adenomas. Tubular adenomas were characterized by reduced microbial methanogenesis and mevalonate metabolism, while serrated adenomas showed increased NAD/NADPH, bile acid, and sulfur metabolism. Many of these microbial and functional shifts were linked to environmental exposures, including diet and common medications such as aspirin. These findings underscore the importance of integrating metagenomic, functional, and environmental data when studying early neoplastic changes in the colon.

Investigations across diverse populations offer additional insights. Recent cohort work from Thailand further underscores this point. Using full- length 16S rRNA sequencing with PICRUSt2, Intarajak et al. [15] profiled stool from patients with hyperplastic polyps and tubular adenomas, reporting lesion-specific microbial and predicted functional signature – hyperplastic polyps enriched for sulfur-oxidation pathways (linked to sulfate-reducing bacteria) and tubular adenomas enriched for mevalonate metabolism, alongside reduced co-occurrence among SCFA producers. These data highlight that early, premalignant lesions can be accompanied by distinct community configurations and metabolic capacities across populations, motivating shotgun metagenomic studies to measure functions directly.

Our study aimed to evaluate the complex interplay between changes in the composition of the gut microbiota and the development of colorectal adenomas, known as precursors of CRC. The primary hypothesis was that the gut microbiota composition differs between individuals with and without colorectal adenomas. We described observed differences in gut microbiota composition and functions between individuals with and without colorectal adenomas.

## Materials and methods

This was a single-centre case-control study conducted by a single expert with an adenoma detection rate (ADR) of 36% in the screening population. The study was conducted from April 1, 2021, to April 22, 2022, at the outpatient endoscopy unit of Health Center 4 in Riga, Latvia, as well as the Academic Histology Laboratory and the Latvian Biomedical Research and Study Center.

## Ethics and informed consent statement

All study procedures were approved by the Central Medical Ethics Committee of Latvia (Permit No. 01–29.1.2/1751) and were conducted in accordance with the Declaration of Helsinki. All participants were adults aged 18 years or older; no minors were enrolled.

Written informed consent was obtained from every participant prior to any study-related procedures. Participants received verbal and written information about the study goals, procedures, risks, data handling, and confidentiality. No waivers of consent were requested or granted.

This was a prospective study involving newly collected stool samples and clinical data; no retrospective data review and no use of anonymized archived samples were performed.

## Participants and study design

In total, 146 patients were recruited for the study; however, ten patients were excluded due to poor bowel preparation, ulcerative colitis, or colorectal cancer. Overall, 136 patients over the age of 18, undergoing colonoscopy for various reasons, who provided informed consent and met the inclusion criteria, were included in the study.

The exclusion criteria were:

- A history of colonoscopy procedures;

- Inflammatory bowel diseases;

- Hereditary polyposis syndrome;

- Established CRC;

- A history of significant intestinal surgeries (intestinal resection, bariatric surgery, etc.), except for appendectomy;

- Any contraindications for polypectomy;

- Incorrect and poor bowel preparation according to Boston Bowel Preparation Scale (BBPS) of 0–1 in any of the three bowel segments;

- Patients with standard contraindications to colonoscopy (including acute diverticulitis/suspected perforation);

- Incomplete colonoscopy procedure (technical difficulties);

- Women who are pregnant or breastfeeding;

- Any acute illness up to the time of inclusion;

- Oncologic diseases diagnosed within the past 3 years or with specific treatment completed less than 3 years ago.

- Chronic kidney disease, autoimmune diseases, HIV, viral hepatitis B or C infections;

- Chronic alcohol consumption;

- Use of antibacterial, probiotic, and immunosuppressive drugs, as well as glucocorticosteroids and proton pump inhibitors in the last two months;

- Diarrhea in the last two weeks.

To minimize the impact of major dietary and medication-related confounders on gut microbiome composition, patients who had used antibiotics or probiotic supplements within one month prior to inclusion were excluded from the study. In addition, none of the included participants reported adherence to a vegan or vegetarian diet at the time

of enrolment. Although detailed dietary intake data were not systematically collected, these criteria were applied to reduce extreme dietary patterns known to influence gut microbial composition substantially.

## Sample collection

Patients were recruited during the consultation, and a colonoscopy examination was scheduled, based on the indicated medical needs. During this consultation, after giving informed consent, patients were given a fecal collection kit. Participants were instructed to collect a stool sample on the day before bowel preparation using a sterile tube provided at the consent visit after giving informed consent. After defecation, samples were kept at room temperature for no longer than 15 minutes and then placed in the participant's home freezer (- 20 °C). At the time of colonoscopy procedure (typically 24 h after sample collection), participants transported the frozen tube to the endoscopy unit. On arrival, samples were immediately placed into a medical −20 °C freezer. Within 48 h of collection, all specimens were transferred to long- term storage at −80 °C in Latvian Genome Centre where they remained until aliquoting for DNA extraction. Only samples collected prior to any bowel preparation and meeting all requirements (room temperature ≤15 min; home freezing initiated ≤15 min post- defecation) were included. During aliquoting, tubes were handled on dry ice, and a single freeze- thaw occurred only at the point of DNA extraction. These procedures are consistent with established recommendations indicating that short- term −20 °C stabilization followed by −80 °C storage preserves overall community profiles for shotgun metagenomics. Although all samples were frozen within 15 minutes of defecation and subsequently stored at −80 °C, some oxygen- sensitive taxa might have been affected during early handling. However, such short- term storage is consistent with prior validated protocols [16].

## Colonoscopy procedure

The examination methodology complied with recognized standards and requirements and was described in detail in the author's previously published work [17]. Colonoscopy examinations were performed with the Olympus EVIS EXERA III (CF-HQ190L/I) video colonoscope. All examinations were performed by a single endoscopist who had 9 years of experience and performs over 1300 colonoscopies annually. In addition, colonoscopy procedures were performed under the supervision of an anesthesiologist, using short-term intravenous sedation based on propofol. The dosage of medication was determined by the anesthesiologist.

Bowel cleanliness was assessed by an endoscopist using the BBPS (Boston Bowel Preparation Scale). Four subjects were excluded from the study due to inadequate bowel preparation, as indicated by a score of 0–1 in any of the three bowel sections. The time of evacuation of the instrument from the cecum for each performed colonoscopy was not less than 7 minutes and was monitored by the endoscopy assistant.

Any detected polyps were described in the colonoscopy report according to the Paris [18] and NICE [19] classifications; the location and size of each polyp in the colon were also specified.

Morphological analysis was performed for all detected or removed polyps. When a polyp was identified, at least two biopsy samples were taken prior to polypectomy. Polyps were resected using either the cold loop or diathermocoagulation technique (hot loop), depending on size. In cases of non-resectable lesions, multiple biopsies were taken, and patients were referred for surgical treatment.

## Morphological diagnosis of lesions found during colonoscopy

All removed polyps and specimens from biopsied lesions were transmitted to the Academic Histology laboratory (Riga, Latvia) for morphological diagnostics. All samples were analyzed by expert pathologists and characterized according to the World Health Organization criteria, depending on morphological characteristics [20]. All lesions were described as serrated polyps and lesions, low-grade dysplasia (LGD), high-grade dysplasia (HGD), superficial submucosal invasive

carcinoma (SM-s; < 1000 µm of submucosal invasion) and deep submucosal invasive carcinoma (SM-d; ≥ 1000 µm of submucosal invasion). No traditional serrated adenoma (TAS), sessile serrated lesion with dysplasia (SSL-D), or unclassified serrated adenoma were found morphologically.

## Determining microbiome composition by metagenome sequencing

Microbial DNA was extracted from faeces provided by study participants using the FastDNA Spin Kit for Soil (MP Biomedicals). The amount of DNA extracted was assessed using the Qubit dsDNA HS Assay Kit reagent kit. The DNA samples were stored at minus 20°C in the restricted-access facilities of the Latvian Genome Centre.

## Metagenome sequencing

Libraries for metagenomic shotgun sequencing were prepared using MGIEasy Universal DNA Library Prep Kit (MGI Tech Co., Ltd.). The input of DNA was 300 ng. Preparation steps briefly: DNA shearing into 420 bp fragments by S220 focused-ultrasonicator (Covaris) followed by size selection using magnetic beads; end repair and A-tailing; adapter ligation followed by magnetic beads cleanup of adapter-ligated DNA; PCR amplification and cleanup of the product; quality control; denaturation; single strand circularization; enzymatic digestion; cleanup of enzymatic digestion product; quality control. The quality and quantity of the resulting libraries was determined with an Agilent Bioanalyzer 2100 and a Qubit® 2.0 fluorometer. Metagenome sequencing was performed using the DNBSEQ-G400RS sequencing platform with the reagent set DNBSEQ-G400RS High-throughput Sequencing Set (FCL PE150) (MGITechCo., Ltd.) according to manufacturer's instructions, obtaining 20 million reads per sample.

## Data analysis

Read quality evaluation was performed with FastQC. Adapter cutting and read trimming was performed with fastp (0.20.0) by using default trimming parameters. Paired reads with a length of 100 bp or longer were retained for further data processing. Reads originating from the host were removed with bowtie2 (v2.3.5.1) using GRCh38 as a reference. Taxonomic classification was performed with Kraken 2.1.2 and Bracken 2.7 against UHGG database (version 2) using Kraken's confidence threshold value of 0.1. HUMAnN 3.8 [21] was used to perform functional profiling.

## Statistical analysis

Statistical analyses were performed using R Studio 4.4.1 [22]. Depth normalization for alpha diversity calculation was performed by constructing a multinomial distribution from metagenomic read count table and drawing n samples from each distribution where n = 101889 is the minimum number of brackens classified reads for sample with the lowest coverage. For each sample alpha diversity indices (Shannon, Simpson, inverse Simpson and Pielou's evenness) were calculated and this process was repeated 10000 times in total. The mean values of the diversity indices across iterations were then used as the representative alpha diversity metrics for each sample. Aitchison's distance was used to evaluate the beta diversity. Transformed taxonomic data obtained by centered log ratio transformation with scikit-bio 0.5.5 were used for the construction of principal component analysis biplot with scikit-learn 0.22.

Differential abundance was tested with limma 3.60.2. Samples with less than 100000 assigned metagenomic reads were removed. Metagenomic features with at least 100 reads present in at least 10% of samples were retained. The P-value of < 0.05 was considered statistically significant. In addition, differential abundance and functional data were analyzed using the R packageMaAsLin2 [23], with adenoma status, sex, BMI, smoking status, and the presence of gastrointestinal diseases as fixed effects, and run ID as a random effect. A default Q-value (FDR) of < 0.25 was considered statistically significant.

This was an exploratory cross-sectional case– control study; therefore, no a priori sample size calculation was performed. The sample size (n = 136) was determined based on recruitment feasibility and consistency with similar metagenomic studies investigating microbiota- adenoma associations [14,24].

## Results

Of the 136 enrolled participants, 123 samples passed quality control and were included in taxonomic analyses, while 135 samples were retained for functional pathway profiling.

### Demographic data

A total of 146 individuals were recruited for this study, including 85 polyp-free individuals, and 61 with colorectal adenomas. During the study, 10 patients were excluded (3 patients with ulcerative colitis, 3 with colorectal cancer, and 4 with poor bowel preparation). All 136 patients were divided into two groups based on the presence (42%, n = 56) or the absence (58%, n = 80) of colorectal adenomas. Patient characteristics are summarized in Table 1.

In total, 77 adenomas were found in 56 patients. From 56 patients 8 had high-risk adenomas (14%) and 48 had low-risk adenomas (86%), p < 0.001.

Out of the adenoma-positive patients, 49% were male (27 patients) and 51% were female (29 patients), with a p-value of 0.05. The male/female ratio of the adenoma and control groups was 27/29 and 26/54, respectively. A total of 51.8% of female patients and 48.2% of male patients had at least one colorectal adenoma. The mean age of adenoma-positive patients was 55.3 ± 13.5, while the mean age of adenoma-negative patients was 45.6 ± 13 (p < 0.001).

Among male patients, 15.4% of all detected colorectal adenomas were high-grade dysplasia (HGD), compared to 11.1% in female patients (p = 0.704). Overall, 41 patients (73.2%) had single tubular adenomas and 15 (26.8%) patients had multiple tubular adenomas (p < 0.001).

The BMI of the adenoma group was 27.51 ± 4.18, while that of the control group was 25.73 ± 5.22 (p = 0.039). Statistically significant differences were found between patients with and without adenomas in age, gender distribution, educational level, place of residence, and physical activity habits (all p < 0.05). Patients in the adenoma group were older, more likely to live in the capital, more likely to have higher education, and significantly more likely to be physically active compared to the control group.

### Microbiome composition

The median number of paired-end reads obtained was 22830377 (IQR 10336229). After quality control and exclusion of host reads, 123 samples from 136 were retained for analysis (50 from adenoma positive and 73 from adenoma-negative

**Table 1. Demographic characteristics of adenoma and control group patients.**

| Variable | All (n = 136) | Adenoma group (n = 56) | Control group (n = 80) | p-value |
|---|---|---|---|---|
| Age (mean ± SD) | | 53.03 ± 5.4 | 45.61 ± 3.0 | <0.001 |
| Gender, N (%) – Male | 53 | 27 | 26 | 0.027 |
| Gender, N (%) – Female | 83 | 29 | 54 | 0.027 |
| BMI (mean ± SD) | | 27.5 ± 4.2 | 25.7 ± 5.2 | 0.039 |
| Place of residence- Capital city | 52 | 32 | 20 | 0.053 |
| Place of residence- Outside capital | 84 | 24 | 60 | 0.053 |
| Education- bachelor's or equivalent | 84 | 47 | 37 | <0.001 |
| Educstion – Lower levels | 52 | 9 | 43 | <0.001 |
| Lifestyle- Physically active | 55 | 41 | 14 | <0.001 |
| Lifestyle- Physically inactive | 81 | 15 | 66 | <0.001 |
| Smoking- Smoker | 8 | 5 | 3 | 0.28 |
| Smoking- Non-smoker | 128 | 51 | 77 | 0.28 |

patients), with a median of 22658664 reads per sample (IQR-interquartile range: 10326052). The median percentage of classified reads was 88.76% (IQR 2.71%).

The relative abundance of the detected genera in patients with colorectal adenomas (AD) and patients without adenomas (CO) is shown in Fig 1. The most common genera in both groups were *Faecalibacterium, Bacteroides, Blautia_A, Alistipes, Prevotella* and *Roseburia*. In addition, the archaeal representative *Methanomassiliicoccus_A* was found to be present among the 20 most common genera.

### Diversity analysis

**Alpha diversity.** Four commonly used diversity metrics were calculated: Shannon index, Simpson index, inverse Simpson index, and Pielou's evenness. Data are presented as mean ± standard deviation (SD). Overall, alpha diversity showed slightly higher diversity in the control group compared to the adenoma group, but none of these differences reached statistical significance.

**Shannon index**: The mean- 4,6 ± 0,42 in the CO group and 4,53 ± 0,43 in the AD group.

**Inverse Simpson index**: The mean- 41,47 ± 16,28 in CO and 37,68 ± 17,37 in AD.

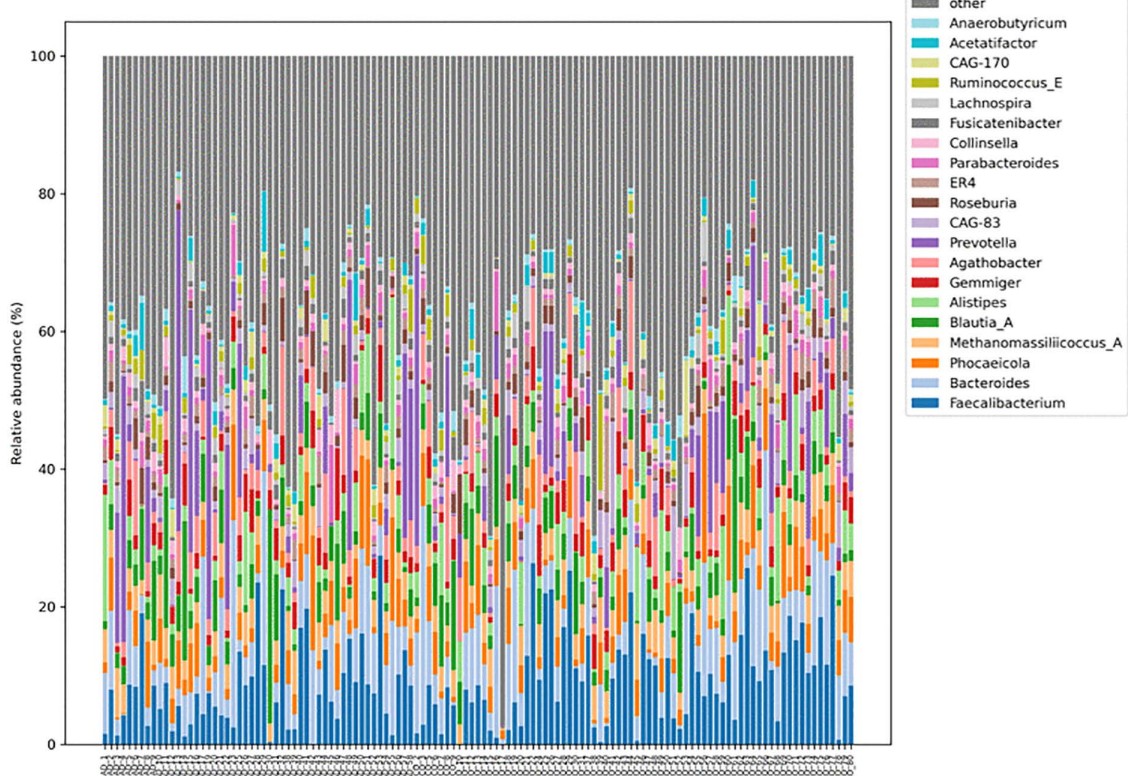

**Fig 1. Genus-level taxonomic composition of fecal microbiota.** Genus-level relative abundance of fecal microbiota in patients with colorectal adenomas (AD) and adenoma-free controls (CO), assessed by shotgun metagenomic sequencing. Bar plots display the relative abundance of the 20 most prevalent bacterial genera across study groups. Both groups were dominated by *Faecalibacterium, Bacteroides, Blautia_A, Alistipes, Prevotella*, and *Roseburia*. Differences shown are descriptive; no genus-level taxa reached statistical significance after false discovery rate (FDR) correction in multivariable differential abundance analyses.

**Pielou's evenness**: The mean- 0,65 ± 0,05 in CO and 0,64 ± 0,05 in AD.

**Simpson index**: The mean-0,97 ± 0.02 in CO and 0,97 ± 0,02 in AD.

These results indicate a trend to higher microbial diversity in patients without colorectal adenomas; however, the observed differences were not statistically significant.

**Beta diversity.** Beta diversity analysis at the species level using PCA (Principal Component Analysis) showed statistically significant differences between AD and CO (p = 0.0002). The principal components PC1 and PC2 together explained 18.37% of the variance, indicating differences in the structure of the microbial community (Fig 2). The AD group had greater microbiota heterogeneity and a stronger association with the genera *Prevotella, Bacteroides, Collinsella* and *Holdemanella*, while the CO group was enriched with *Faecalibacterium*, *Anaerostipes* and other beneficial taxa. CO samples clustered more tightly, indicating a more stable microbiome.

### Differential abundance analysis

MaAsLin2 models were adjusted for adenoma status, age, sex, BMI, smoking status, gastrointestinal comorbidities, and run ID. Differential abundance analysis using MaAsLin2 with FDR correction (<0.25) identified one statistically significant altered taxon- *UBA7597 sp003448195.*

This microorganism showed a significantly higher relative abundance in the AD group compared to the CO group (LogFC = 3.44; FDR = 0.002). A positive LogFC value corresponds to an increase in the AD group. Although the mean relative abundance was low, the result reached statistical significance (p = $1.03 \times 10^{-6}$), and the Bayes factor (B = 4.68) supported the robustness of this finding.

The only species-level feature passing FDR correction was UBA*7597 sp003448195*, a UHGG v2 placeholder metagenome- assembled genome (MAG) taxonomically classified within *Firmicutes_A> Clostridia_A> Oscillospirales> Oscillospiraceae* [25,26]. Because this genome lacks a cultured representative and a curated phenotype, it should be considered taxonomically uncharacterized.

In HUMAnN species-stratified pathway tables, *UBA7597 sp003448195* contributed negligibly to each of the six pathways that differed between groups, indicating that the observed functional differences reflect community-level metabolic alterations rather than being attributable to this taxon alone.

No other taxa were found to have significant differences between groups after FDR correction. Results were obtained by analyzing 123 metagenomic samples using the linear model approach (limma) and MaAsLin2 (Fig 3).

### Functional analysis

To investigate the potential differences in microbial functions depending on colorectal adenoma presence status, we evaluated the functional profile of the gut microbiome. Multivariable association analysis considering various fixed effects, revealed significant differences in six functions depending on colorectal adenoma presence using the MaAsLin2 model with an FDR < 0.25. The results of the analysis have been summarized in Fig 4. Of the 136 participants enrolled in the study, 135 samples were retained for functional pathway analysis after quality control.

All pathways showed a higher relative abundance in the CO group. Two of them- PWY-7371 and PWY-7992 were associated with menaquinone (vitamin K2) biosynthesis (coefficients 1.57 and 1.52; FDR = 0.14). Three pathways – PROPFERM-PWY, PWY-8188 and PWY-8189 – corresponded to Stickland fermentation pathways with oxidative and reductive reactions of amino acids (each with a coefficient of 1.04; FDR = 0.18). In addition to PWY-5494 (pyruvate fermentation to propanoate II), the pathway that contributes to the production of SCFA (propionate) was also dysregulated in the AD group (coefficient 1.01; FDR = 0.23). All identified pathways were more widely distributed and had higher mean values in the CO group, while their reduced abundance and narrower distribution were observed in the AD group. Results were obtained by analyzing 135 metagenomic samples, and each of the pathways demonstrated differences between groups.

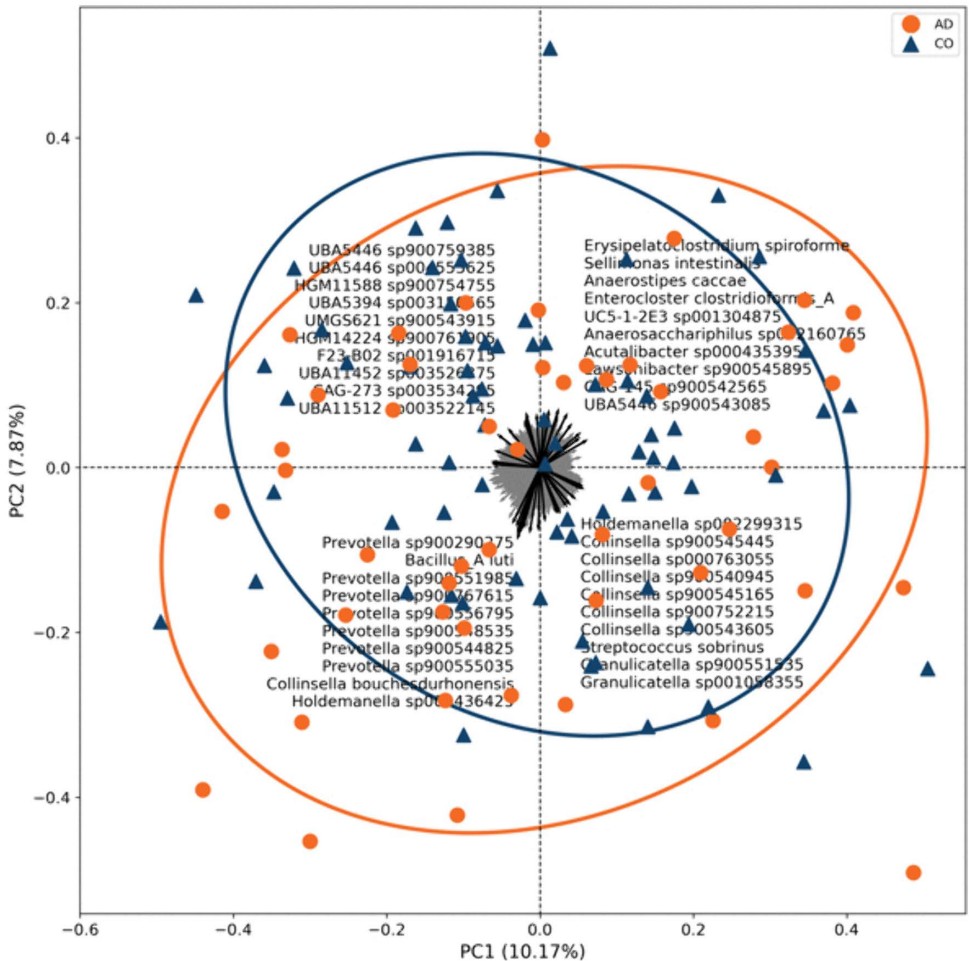

**Fig 2. Beta diversity analysis of gut microbiota.** Principal component analysis (PCA) of gut microbiota beta diversity at the species level in adenoma-positive (AD, n = 50) and adenoma-negative control (CO, n = 73) groups, based on Aitchison distance after centered log-ratio (CLR) transformation. Each point represents an individual sample. The first two principal components (PC1 and PC2) together explain 18.37% of the total variance. Overall microbial community structure differed significantly between groups (p = 0.0002), with greater inter-individual variability observed in the adenoma group.

## Discussion

CRC is a major global health burden. Gut microbiota alterations (dysbiosis) have been linked to colorectal adenomas – the precursors of CRC [27].

Yet, despite prior work [11–13], evidence on microbiome differences between individuals with and without adenomas remains inconclusive, warranting further study.

We analyzed gut microbiota composition in 136 patients (123 after QC) undergoing colonoscopy, grouped into AD (adenoma) and CO (control) groups. Only one microbial taxon differed significantly between groups. Most detected adenomas were low-risk tubular adenomas <10 mm without high-grade dysplasia or villous structure – a category rarely investigated in previous studies [28].

Colorectal adenoma incidence in our study appeared influenced by host and environmental factors affecting gut microbiota. Advanced age is linked to microbiota shifts toward a pro-inflammatory and less functionally diverse profile, potentially increasing neoplastic risk [29–32]. Sex-related hormonal and immune differences may influence gut microbiota and

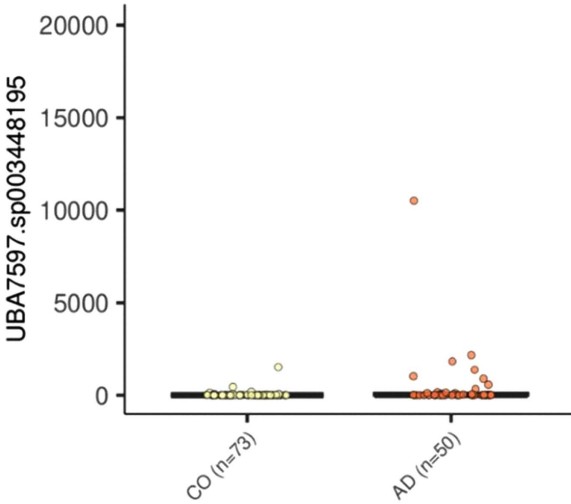

**Fig 3. Differential abundance of *UBA7597 sp003448195* in adenoma and control groups.** Differential abundance of *UBA7597 sp003448195* between adenoma-positive (AD, n = 50) and control (CO, n = 73) groups identified using MaAsLin2 multivariable modeling. Models were adjusted for adenoma status, age, sex, BMI, smoking status, gastrointestinal comorbidities, and sequencing run ID. The species showed significantly higher relative abundance in the adenoma group (LogFC = 3.44; p = $1.03 \times 10^{-6}$; FDR = 0.002). Boxes represent interquartile ranges, horizontal lines indicate medians, and whiskers denote data range.

colorectal neoplasia risk [33,34]. Lifestyle factors such as urban living, higher education, and smoking can alter microbial balance through lower fiber intake, processed food consumption, and reduced SCFA-producing bacteria [35–38].

Previous studies on gut microbiota diversity in colorectal cancer have shown inconsistent results- some report reduced diversity, others no significant differences. Our findings align with those suggesting that alpha diversity alone is not a reliable marker of adenoma presence [39,40].

While alpha diversity did not differ significantly between groups, beta diversity analysis demonstrated differences in overall microbial community structure, indicating altered microbiome composition in adenoma-positive patients. PCA patterns were descriptive and should not be over-interpreted. Similar findings were reported by Deng et al. [24], who as well observed increased inter-individual variability despite unchanged overall diversity [4].

In this study, beta diversity did not differ significantly between AD and CO. Descriptively, AD showed greater inter-individual variability on the PCA biplot. The genus- level composition was dominated by *Faecalibacterium*, *Bacteroides*, *Blautia_A*, *Alistipes*, and *Prevotella*, while differential abundance analysis identified only one statistically significant species-level feature- *UBA7597 sp003448195* (LogFC = 3.44; FDR = 0.002). Members of the *Oscillospiraceae* family are obligate anaerobes involved in energy recovery from dietary fiber fermentation and are often associated with butyrate and propionate production, which contribute to epithelial homeostasis and immune regulation [26].

Functional profiling revealed reduced microbial pathways for menaquinone (vitamin $K_2$) biosynthesis, Stickland fermentation, and short-chain fatty acid (propionate) production in the AD group, suggesting community-wide metabolic alterations linked to early dysbiosis.

Genera such as *Prevotella*, *Collinsella*, and *Holdemanella* appeared more frequently in the AD group, though differences were not statistically significant. *Prevotella* has been linked to mucosal inflammation and low-fiber diets [41], while *Faecalibacterium*, *Anaerostipes*, and *Bacteroides* produce butyrate, supporting mucosal integrity and anti-inflammatory balance [42–44]. These trends align with previous studies suggesting that shifts between pro- and anti-inflammatory taxa may accompany, rather than cause, early adenoma- related changes [45–48].

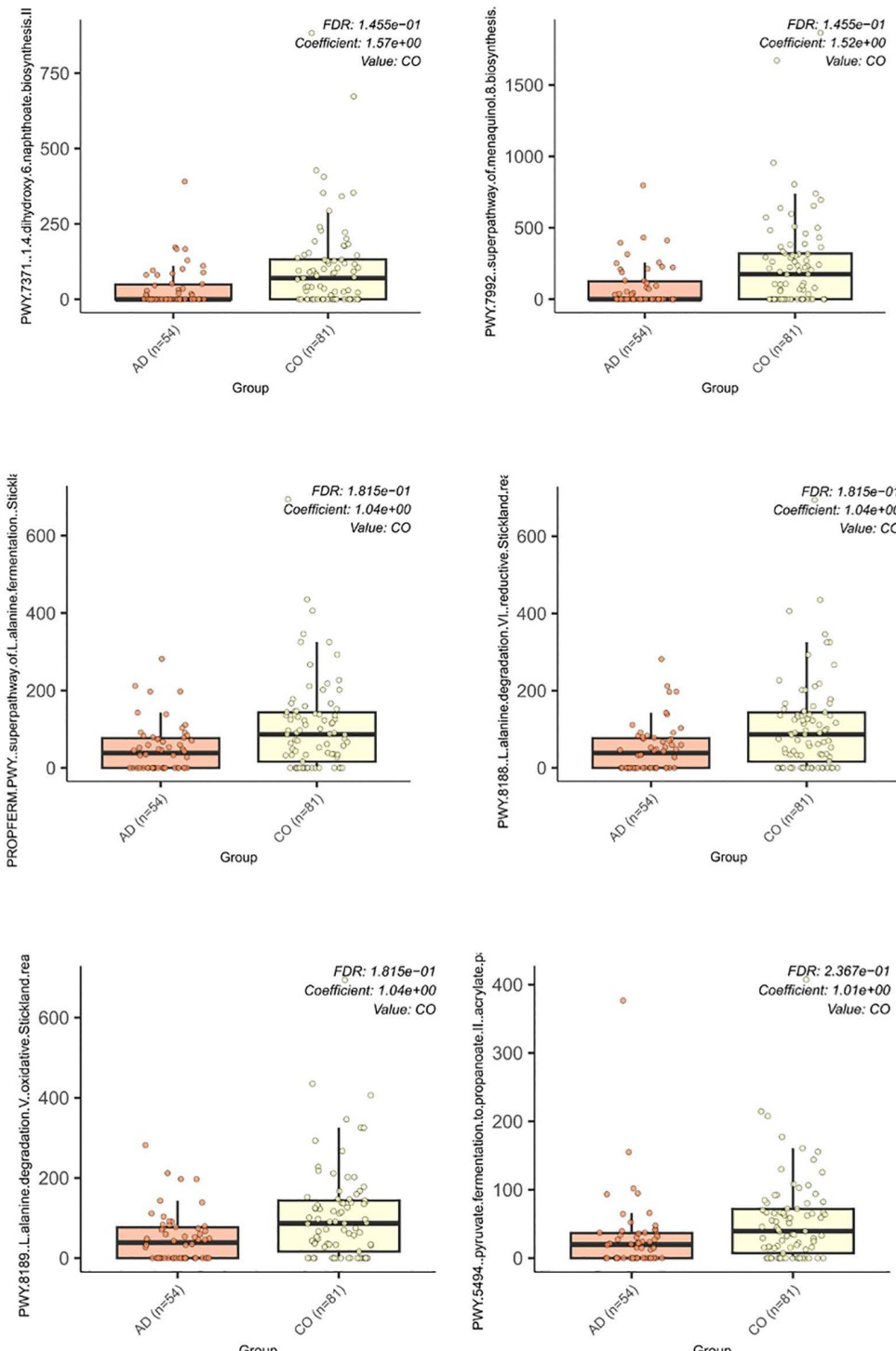

**Fig 4. Differentially abundant microbial functional pathways.** Differentially abundant microbial metabolic pathways between adenoma-positive (AD) and adenoma-negative control (CO) groups identified using HUMAnN functional profiling and MaAsLin2 multivariable analysis. Pathways shown met the significance threshold of FDR < 0.25. All displayed pathways were more abundant in control samples and included menaquinone (vitamin $K_2$) biosynthesis pathways (PWY-7371, PWY-7992), Stickland amino acid fermentation pathways (PROPFERM-PWY, PWY-8188, PWY-8189), and pyruvate fermentation to propionate (PWY-5494). Results reflect community-level functional differences rather than contributions of individual taxa.

*Collinsella* is also known to affect intestinal permeability and bile acid metabolism and has been associated with metabolic disorders and low-grade inflammation [47,49].

Differential abundance analysis identified one significant taxon - *UBA7597 sp003448195*- enriched in adenoma patients. According to the UHGG v2 database, it belongs to *Firmicutes_A > Clostridia_A > Oscillospirales > Oscillospiraceae* and represents an uncultured metagenome-assembled genome [25]. While its phenotype remains uncharacterized, genomic evidence from related *Oscillospiraceae* members suggests involvement in anaerobic carbohydrate fermentation and short-chain fatty acid production- processes essential for colonic energy balance and mucosal homeostasis.

The higher abundance of *UBA7597 sp003448195* in the AD group should be viewed as an association rather than causation. This uncharacterized *Oscillospiraceae* genome contributed minimally to altered pathways in HUMAnN, suggesting community- level rather than taxon-specific effects. Despite strong statistical significance (LogFC = 3.44; FDR = 0.002; p = $1.03 \times 10^{-6}$; B = 4.68), it should be considered a potential biomarker of microbiome alteration, not a driver of pathology. Similar findings were reported by Deng et al. [24], who observed higher abundance of *Esherichia, Shigella* and *Clostridium* species in polyp patients, indicating microbial shifts may mark early dysbiosis in adenoma formation.

To complement taxonomic findings, we analyzed functional pathway differences between groups. Six pathways (all FDR < 0.25) showed higher activity in controls, including two for menaquinone biosynthesis (PWY-7371, PWY-7992), three for Stickland fermentation (PWY-8188, PWY-8189, PROPFERM-PWY), and one for SCFA production (PWY-5494).

Literature identifies model contributors to these pathways, including *Escherichia coli* [49,50], *Bacteroides fragilis* [51], *Clostridium spp., Klebsiella aerogenes, Bacillus subtilis*, and certain *Lactobacillus* species [33,52], which are known to support redox balance and gut metabolic function.

Reduced menaquinone biosynthetic activity in adenoma patients likely reflects altered microbial metabolic potential rather than causation. While vitamin $K_2$ modulates epithelial proliferation and oxidative balance [51,53–55], our results show only an association between decreased pathway abundance and adenoma presence. Gut microorganisms such as *Lactococcus*, *Bacteroides*, and *Eubacterium*, key menaquinone producers, are influenced by diet and inflammation [56]. Beyond microbial metabolism, vitamin $K_2$ also serves as an essential cofactor in host cellular processes, including γ-carboxylation of proteins involved in cell growth regulation, apoptosis, and immune modulation. Experimental studies have shown that menaquinone can suppress colorectal cancer cell proliferation and promote apoptosis through modulation of mitochondrial electron transport and NF-κB/MAPK signaling pathways [57,58]. Therefore, reduced microbial menaquinone biosynthetic capacity may be consistent with altered epithelial redox balance and mucosal immune homeostasis; however, host-level measurements are needed.

Previous findings [24,56,59] similarly reported early microbiome functional changes in tubular adenomas, suggesting such shifts may act as biomarkers of early mucosal dysbiosis.

Adenoma patients showed reduced abundance of Stickland fermentation pathways- key for anaerobic amino acid metabolism and nitrogen cycling [60] – and decreased activity of PWY-5494 (pyruvate fermentation to propionate II), involved in SCFA synthesis. Since SCFAs like propionate and butyrate support epithelial integrity and immune balance [61,62], these reductions suggest loss of beneficial microbial functions. Overall, adenoma presence was linked to lower activity in menaquinone, Stickland, and SCFA pathways, reflecting altered metabolic homeostasis; however, these should be interpreted as associations, not causal mechanisms.

Reduced menaquinone biosynthetic activity may indicate fewer bacteria capable of vitamin $K_2$ synthesis, which supports epithelial stability and has anti-proliferative effects on CRC cells. However, this shift likely reflects association rather than causation. Likewise, reduced Stickland fermentation and SCFA biosynthesis – especially propionate production suggests altered amino acid and energy metabolism, with potential impacts on barrier integrity and inflammation regulation, but these patterns should be viewed as microbial imbalances rather than direct tumorigenic drivers.

These associations illustrate the complex ecological and metabolic remodeling of the gut microbiome accompanying colorectal adenomas. Our metagenomic data showing reduced menaquinone, Stickland, and SCFA pathways align with findings from other cohorts suggesting that early neoplasia involves loss of beneficial metabolic capacity

Similarly, Intarajak et al. [15] reported reduced SCFA producers and altered mevalonate and sulfur metabolism in Thai patients with polyps, supporting the view that such shifts reflect early dysbiosis rather than causation. Longitudinal and experimental studies are needed to determine whether these changes precede or follow adenoma formation.

Our findings support that colorectal adenoma development is accompanied by subtle but functionally relevant microbiome shifts, consistent with international studies [14]. Decreased activity of pathways for vitamin $K_2$, propionate, and amino acid fermentation suggests loss of commensal functions linked to barrier integrity and anti-inflammatory balance. Despite a modest sample size, our Eastern European metagenomic data add geographic diversity to current evidence. Future longitudinal studies integrating dietary data and probiotic or dietary interventions could clarify whether these shifts precede adenoma formation or represent adaptive responses.

Study limitations include a relatively young cohort, modest sample size, and lack of cross-population comparison. Detailed dietary data were unavailable, which may have influenced pathways related to menaquinone and SCFA (propionate) production. Despite adjusting MaAsLin2 models for major covariates (adenoma status, age, sex, BMI, smoking status, gastrointestinal comorbidities, and run ID), residual confounding by demographic and lifestyle factors such as education, residence, and physical activity cannot be fully excluded. Notably, the AD group exhibited higher physical activity and education levels, a finding that appears counterintuitive given their typically protective roles. This pattern likely reflects a selection bias inherent to screening-based recruitment, as individuals with higher education and health awareness are more likely to participate in preventive colonoscopy programs and report healthier lifestyles. Thus, these variables may not indicate causal associations but rather differences in health-conscious behavior and participation patterns

These variables are highly interrelated and may partially underlie the observed microbiota and functional differences, including the abundance of *UBA7597 sp003448195* and altered metabolic pathways. Therefore, our findings should be interpreted as associations rather than causative effects. Future studies with larger, more homogeneous populations and harmonized dietary and lifestyle data are essential to disentangle these interdependent influences and confirm the biological relevance of the observed associations. Additionally, the cross sectional design of this study precludes conclusions about causality and does not allow determination of whether microbiota alterations precede or result from adenoma formation. Long-term dietary and lifestyle factors, such as fiber intake and alcohol consumption, were not quantitatively evaluated and could have influenced microbiome composition and function.

Furthermore, functional inferences were not validated at the metabolite level, limiting direct confirmation of the observed pathways' physiological impact. Future research integrating host indicators such as serum vitamin $K_2$ levels, inflammatory markers, and targeted fecal metabolomics would enable validation of microbial functional shifts and provide a more complete understanding of the gut microbiome's role in adenoma risk. Although adenoma characteristics such as size, Paris/NICE classification, en bloc resection status, and biopsy number were recorded, they were not included in statistical analysis due to the limited number of high-risk adenomas (n = 8). This small subgroup did not provide sufficient power for meaningful comparison with low-risk lesions (n = 48). Future studies with larger and histologically diverse adenoma cohorts should examine whether microbial and functional alterations display a gradient corresponding to adenoma size or dysplasia severity.

## Conclusions

The identification of microbial taxa and functional pathways linked to adenoma presence highlights the potential for using microbiota-based markers and treatments to prevent and manage colorectal cancer. Future research should focus on establishing connections and clarifying how gut microbiota influences colorectal carcinogenesis.

## Acknowledgments

We thank the Genome Database of the Latvian Population for their support in processing the biological samples and related data used in this study.

## Author contributions

**Conceptualization:** Ilona Vilkoite.

**Methodology:** Jānis Kloviņš, Ivars Tolmanis, Aivars Lejnieks.

**Supervision:** Aivars Lejnieks.

**Visualization:** Ivars Silamiķelis, Elīna Runce, Krista Cēbere, Ksenija Margole.

**Writing – original draft:** Laila Silamiķele.

**Writing – review & editing:** Olga Sjomina.

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
