## [Decision Letter · Decision Letter 0]

20 Jan 2026

Dear Dr. Vilkoite,

Thank you for submitting your manuscript to PLOS ONE. After careful consideration, we feel that it has merit but does not fully meet PLOS ONE’s publication criteria as it currently stands. Therefore, we invite you to submit a revised version of the manuscript that addresses the points raised during the review process.

We look forward to receiving your revised manuscript.

Kind regards,

Jad El Masri

Academic Editor

PLOS One

Journal Requirements:

“This work was funded by the European Regional Development Fund (ERDF), Measure 1.1.1.1 “Support for applied research” Project No.1.1.1.1/18/A/092 “Role of miRNAs in the host-gut microbiome communication during metformin treatment in the context of metabolic disorders”.”

3. Please note that your Data Availability Statement is currently missing the DOI/accession number of each dataset OR a direct link to access each database. If your manuscript is accepted for publication, you will be asked to provide these details on a very short timeline. We therefore suggest that you provide this information now, though we will not hold up the peer review process if you are unable.

Reviewers' comments:

Reviewer's Responses to Questions

**Comments to the Author**

1. Is the manuscript technically sound, and do the data support the conclusions?

Reviewer #1: Yes

Reviewer #2: Yes

Reviewer #3: Yes

2. Has the statistical analysis been performed appropriately and rigorously?

Reviewer #1: Yes

Reviewer #2: Yes

Reviewer #3: Yes

3. Have the authors made all data underlying the findings in their manuscript fully available?

Reviewer #1: Yes

Reviewer #2: Yes

Reviewer #3: Yes

4. Is the manuscript presented in an intelligible fashion and written in standard English?

Reviewer #1: Yes

Reviewer #2: Yes

Reviewer #3: Yes

Reviewer #1: This paper analysis gut microbiome composition and function to colorectal adenoma using shotgun metagenomics. Cēbere et al. recruited 135 patients among patient receiving standard colonoscopy between April 1, 2021 and April 22, 2022 based on strict inclusion and exclusion criteria. Stool samples were collected before colonoscopy for metagenomic shotgun sequencing and were assessed for alpha diversity (taxa diversity within each sample) and beta diversity (taxa diversity between conditions). No significant Alpha diversity was observed between the samples but a Beta diversity was noted. Prevotella, Bacteroides, Collinsella and Holdemanella were more associated with the adenoma group while Faecalibacterium, Anaerostipes were associated with the control group. However, after adjusting for confounding variables only one statistically significant altered taxon- UBA7597 sp003448195 was identified.

Overall, the study was well conducted. The introduction set up well the research background and describe the research problem with the benefit that this work has. Study design was well appropriate with good inclusion and exclusion criteria. Statistical methods used are well suited to examine intra and inter samples differences and remove confounding variables. This paper should be considered for publication with just minor adjustments. In the abstract the authors have stated twice that beta diversity is significantly different (Line 39 and 42), instead they should avoid redundancy. Another issue I found was with the results when reporting the number of samples used. The authors stated that 136 patients were included and After QC, 123 samples retained for taxonomic analysis. But Later in Functional analysis the author stated again using 136 samples “Of the 136 participants included in the overall taxonomic analysis, one sample was excluded from the functional analysis (n = 135).” Sample numbers are confusing and must be fixed.

Reviewer #2: This manuscript titled “Colorectal adenoma presence is associated with decreased menaquinone pathway functions in the gut microbiome of patients undergoing routine colonoscopy is a prospective, case-control study using shotgun metagenomics sequencing.

Modifications:

-Abstract should clearly state the type of study design (cross-sectional case-control study) as in this case; this would make it easier for the readers to know what to expect.

-Redundancy: it was mentioned twice “Beta diversity analysis showed statistically significant differences” (lines 39-40 and lines 42-43).

-in the introduction: Line 86–90: “Although various studies have explored changes in gut microbiota composition in the context of colorectal adenomas, the results remain inconclusive…” It would be helpful to briefly specify in what way they are inconclusive.

- in the methods section, recommended to add the dietary and medication data; this might affect the results

- results section: There is a discrepancy between results and the way they are reported. In the Abstract, Bacteroides and Prevotella are mentioned to be increased but the in the results after proper adjustment and FDR correction Only UBA7597 sp003448195 is significant

Recommendation: Minor Revision

Reviewer #3: The question addressed by the authors is very important, and the manuscript presents valuable real-world data that contributes to our understanding of the gut microbiome composition in relation to colorectal adenoma. However, some minor edits are prompted:

1. Beta diversity contradiction: in the discussion the author states that there is no statistical beta diversity “While alpha and beta diversity did not differ significantly between groups…” (lines 390–391) however beta diversity was significant in the results: “Beta diversity analysis at the species level … showed statistically significant differences between AD and CO (p = 0.0002).”

2. In this sentence “Changes in gut microbiota appear to be linked to CRC by promoting chronic inflammation, immune dysfunction, and metabolic issues…” “Metabolic issues” is vague; try changing the term or giving a precise alteration.

3. The figures are presented with insufficient legends. Please add clear descriptions.

4. The conclusion in the abstract is very broad and does not tell me anything

5. The background in the abstract is weak and too general

**Do you want your identity to be public for this peer review?** For information about this choice, including consent withdrawal, please see our Privacy Policy

Reviewer #1: No

Reviewer #2: No

Reviewer #3: No

---

## [Author Response · Author response to Decision Letter 1]

3 Feb 2026

Dear Jad El Masri,

We sincerely thank you for the careful evaluation of our manuscript and for the opportunity to submit a revised version. We appreciate the thoughtful and constructive comments provided by the academic editor and reviewers, which were very helpful in improving the clarity and presentation of our work. Below, we respond to each of the editorial requirements in detail.

1. PLOS ONE style requirements

The manuscript has been revised to comply with PLOS ONE formatting and style guidelines. File naming and manuscript structure now follow the provided PLOS ONE templates.

2. Role of the funder

The following statement has been included in the cover letter and manuscript: “The funders had no role in study design, data collection and analysis, decision to publish, or preparation of the manuscript.”

3. Data Availability Statement (DOI / accession numbers)

We thank the editor for highlighting this requirement. The Data Availability Statement has been updated to include the repository name, accession number, and a direct access link. All metagenomic sequencing data generated in this study are publicly available in the European Nucleotide Archive (ENA) under accession number PRJEB79034 (https://www.ebi.ac.uk/ena/browser/view/PRJEB79034).

4. Reviewer-recommended citations

Reviewers did not recommend to add any other citation.

5. Reference list

The reference list has been reviewed for accuracy and completeness. No retracted articles are cited. Any updates to the references have been noted in the responses to the reviewers.

Further, we provide our response to the reviewers.

Response to Reviewer #1

We thank the reviewer for the positive evaluation of our study and for the thoughtful and constructive comments. We are pleased that the reviewer found the study design, statistical approach, and overall presentation appropriate, and we appreciate the helpful suggestions for improving clarity.

Regarding the sample numbers, we agree that the original wording may have caused confusion. The total number of enrolled participants was 136. After quality control, 123 samples were retained for taxonomic profiling, whereas functional pathway analysis was performed on 135 samples, as functional profiling has different quality and coverage requirements. We have revised the relevant sections of the Results to clearly distinguish between the number of enrolled participants and the number of samples included in each analysis, thereby improving clarity and consistency throughout the manuscript.

In addition, we have revised the Abstract to remove the redundant mention of statistically significant beta diversity differences, as suggested by the reviewer.

We thank the reviewer again for these valuable comments, which have helped us to improve the clarity of the manuscript.

Response to Reviewer #2

We thank the reviewer for the careful reading of our manuscript and for the constructive and helpful suggestions. We appreciate the opportunity to clarify and improve several aspects of the study. Our responses to each comment are provided below.

1. Study design in the Abstract

We thank the reviewer for this suggestion. The Abstract has been revised to explicitly state the study design as a cross-sectional case–control study, which we agree will help readers more clearly understand the study framework.

2. Redundancy in reporting beta diversity results

We thank the reviewer for pointing out this redundancy. The Abstract has been revised to remove the repeated statement regarding statistically significant beta diversity differences, improving clarity and conciseness.

3. Clarification of inconclusive findings in previous studies (Introduction, lines 86–90)

We thank the reviewer for this helpful comment. We have revised the Introduction to clarify in what way previous findings are inconclusive. Specifically, we now note that published studies differ in the reported direction and magnitude of microbial diversity changes, as well as in the identification of taxa associated with adenoma presence. In addition, methodological heterogeneity - such as differences in sequencing platforms, analytical pipelines, lesion characteristics, and population backgrounds - has contributed to inconsistent and sometimes conflicting results.

4. Dietary and medication data (Methods section)

We thank the reviewer for raising this important point. To minimize the impact of major dietary- and medication-related confounders on gut microbiome composition, patients who had used antibiotics or probiotic supplements within one month prior to inclusion were excluded from the study. In addition, none of the included participants reported adherence to a vegan or vegetarian diet at the time of enrolment. Although detailed dietary intake data were not systematically collected, these criteria were applied to reduce extreme dietary patterns and medication exposures known to influence gut microbial composition substantially. The Methods section has been revised accordingly to clarify these points.

5. Consistency between Abstract and Results regarding significant taxa

We thank the reviewer for highlighting this issue. The Abstract has been revised to clarify that the reported genus-level differences (e.g., Bacteroides and Prevotella) represent descriptive trends observed prior to multiple-testing correction. We now clearly distinguish these descriptive patterns from the species-level analysis, in which only UBA7597 sp003448195 remained statistically significant after multivariable adjustment and FDR correction. This revision ensures full consistency between the Abstract and the Results section.

We thank the reviewer again for the thoughtful and constructive feedback, which has helped us to improve the clarity of the manuscript.

Response to Reviewer #3

We thank the reviewer for the positive assessment of our manuscript and for recognizing the importance of the research question and the value of the real-world data presented. We also appreciate the constructive suggestions, which have helped us to improve the clarity and precision of the manuscript. Our detailed responses are provided below.

1. Contradiction regarding beta diversity

We thank the reviewer for identifying this inconsistency. The statement in the Discussion section has been corrected. While alpha diversity did not differ significantly between adenoma and control groups, beta diversity analysis demonstrated statistically significant differences in overall microbial community structure, as reported in the Results section (p = 0.0002). The revised Discussion text now accurately reflects these findings and is fully consistent with the Results.

2. Clarification of the term “metabolic issues”

We thank the reviewer for this helpful suggestion. The term “metabolic issues” was indeed too vague and has been revised. We now specify that changes in gut microbiota may contribute to colorectal cancer through altered microbial metabolic functions, in addition to chronic inflammation and immune dysfunction. This revised wording more precisely reflects the mechanisms discussed and aligns with the functional results of our study.

3. Insufficient figure legends

We thank the reviewer for highlighting this issue. All figure legends have been revised and expanded to provide clear, self-contained descriptions. The updated legends now specify the analysis performed, sample groups and sizes, statistical approaches, and key observations for each figure, thereby improving interpretability and readability.

4. Broad conclusion in the Abstract

We thank the reviewer for this comment. The Conclusions section of the Abstract has been revised to provide a more specific and data-driven summary of the main findings, highlighting the observed functional differences in microbial metabolic pathways associated with colorectal adenomas.

5. Weak and overly general background in the Abstract

We appreciate this suggestion. The Background section of the Abstract has been strengthened to better emphasize the clinical relevance of colorectal adenomas as early neoplastic lesions and to clearly outline the existing knowledge gap regarding microbiome functional alterations at the adenoma stage, which this study aims to address.

We thank the reviewer again for the insightful ad constructive feedback, which has significantly improved the clarity, consistency, and overall quality of the manuscript.

---

## [Decision Letter · Decision Letter 1]

16 Feb 2026

Colorectal adenoma presence is associated with decreased menaquinone pathway functions in the gut microbiome of patients undergoing routine colonoscopy

PONE-D-25-58757R1

Dear Dr. Vilkoite,

We’re pleased to inform you that your manuscript has been judged scientifically suitable for publication and will be formally accepted for publication once it meets all outstanding technical requirements.

Kind regards,

Jad El Masri

Academic Editor

PLOS One

Additional Editor Comments (optional):

Reviewers' comments:

Reviewer's Responses to Questions

**Comments to the Author**

Reviewer #1: All comments have been addressed

Reviewer #3: All comments have been addressed

2. Is the manuscript technically sound, and do the data support the conclusions?

Reviewer #1: Yes

Reviewer #3: Yes

3. Has the statistical analysis been performed appropriately and rigorously?

Reviewer #1: Yes

Reviewer #3: Yes

4. Have the authors made all data underlying the findings in their manuscript fully available?

Reviewer #1: Yes

Reviewer #3: Yes

5. Is the manuscript presented in an intelligible fashion and written in standard English?

Reviewer #1: Yes

Reviewer #3: Yes

Reviewer #1: (No Response)

Reviewer #3: The authors have implemented all the comments given by the reviewers, and the manuscript has a much better flow.

**Do you want your identity to be public for this peer review?** For information about this choice, including consent withdrawal, please see our Privacy Policy

Reviewer #1: No

Reviewer #3: No

---

## [Editor Report · Acceptance letter]

PONE-D-25-58757R1

PLOS One

Dear Dr. Vilkoite,

I'm pleased to inform you that your manuscript has been deemed suitable for publication in PLOS One. Congratulations! Your manuscript is now being handed over to our production team.

Kind regards,

on behalf of

Dr. Jad El Masri

Academic Editor

PLOS One